# State-of-the-Art Organ-on-Chip Models and Designs for Medical Applications: A Systematic Review

**DOI:** 10.3390/biomimetics10080524

**Published:** 2025-08-11

**Authors:** Gustavo Adolfo Marcelino de Almeida Nunes, Ana Karoline Almeida da Silva, Rafael Mendes Faria, Klériston Silva Santos, Arthur da Costa Aguiar, Lindemberg Barreto Mota da Costa, Glécia Virgolino da Silva Luz, Marcella Lemos Brettas Carneiro, Mário Fabrício Fleury Rosa, Graziella Anselmo Joanitti, Karoany Maria Ibiapina, Ana Karen Gonçalves de Barros Gomes, Adson Ferreira da Rocha, Suélia de Siqueira Rodrigues Fleury Rosa

**Affiliations:** 1Postgraduate Programme in Mechatronic Systems, Department of Mechanical Engineering, Campus Darcy Ribeiro, University of Brasilia, Brasilia 70910-900, DF, Brazil; anakaroline.alms@gmail.com (A.K.A.d.S.); rafaelmendes@iftm.edu.br (R.M.F.); kleristonsantos@iftm.edu.br (K.S.S.); arthuraguiar.eng@gmail.com (A.d.C.A.); suelia@unb.br (S.d.S.R.F.R.); 2Department of Electrical Engineering, Federal Institute of Education, Science and Technology of Triângulo Mineiro, Paracatu 38603-402, MG, Brazil; 3Postgraduate Programme in Biomedical Engineering, Department of Eletronic Engineering, Faculty of Science and Engineering Technologies (FCTE), University of Brasilia, Gama 72444-240, DF, Brazil; bergbmota1515@gmail.com (L.B.M.d.C.); gleciavs@gmail.com (G.V.d.S.L.); marbretas@gmail.com (M.L.B.C.); mariorosafleury@gmail.com (M.F.F.R.); gjoanitti@gmail.com (G.A.J.); adson@unb.br (A.F.d.R.); 4Research and Innovation Center in Organ-on-a-Chip at the University of Brasilia, University of Brasilia, Brasilia 70910-900, DF, Brazil; karoanymaria@gmail.com (K.M.I.); akarengb23@gmail.com (A.K.G.d.B.G.); 5Master of Engineering (M.Eng.) Program, Meinig School of Biomedical Engineering, Cornell University, Ithaca, NY 14853, USA

**Keywords:** Organ-on-a-Chip, microfluidics, microphysiological systems, 3D bioprinting, Patient-Specific Models, cell microenvironment, translational medicine

## Abstract

Organ-on-a-chip (OoC) devices simulate human organs within a microenvironment, potentially surpassing traditional preclinical methods and paving the way for innovative treatments. A thorough understanding of the current state of OoC design enables the development of more precise and relevant models that replicate not only the structure of organs but also their intricate cellular interactions and responses to external stimuli. This knowledge facilitates the optimization of biomimetic materials and allows for the real-time simulation of physiological microenvironments. By keeping abreast of new microfabrication techniques, we can explore opportunities to create customized and highly functional OoCs. Objective: To provide a comprehensive overview of microphysiological platform designs. Methods: This systematic review was registered in PROSPERO under the number CRD42022352569. We adhered to the Preferred Reporting Items for Systematic Reviews and Meta-Analyses (PRISMA) guidelines. The eligibility criteria included studies utilizing human tissue, either primary or secondary lineage cells. Results: A total of 9.790 papers were retrieved from the Scopus, Embase, IEEE and Web of Science databases. After removing duplicates and applying a 10-year publication filter, 3.150 articles were screened by title and abstract. Full-text analyses were then performed. Eighteen studies met the eligibility criteria and were included in this systematic review. In this review, we examine the principles of OoC design, focusing on structure, dimensions, cell culturing options and manufacturing techniques. We also examine recent advances and future prospects in the field. Conclusions: Microphysiological devices in health research can facilitate drug discovery and improve our understanding of human physiology. They contribute to more ethical research by reducing the number of animals used in experiments.

## 1. Introduction

Organ-on-a-chip (OoC) devices play a fundamental role in the field of biomedical engineering, revolutionizing the way we conduct assays and tests related to medical devices and pharmacological studies [1,2,3]. Their importance lies in their ability to accurately replicate the physiological microenvironments of human organs on microscopic scales [4]. This allows for more accurate preclinical assays than traditional methods, such as 2D cell cultures or animal studies, which often do not fully represent the complexity of the human body’s responses to various treatments [2,3,4,5].

The use of OoCs can help minimize the time in the biomedical device development process and shorten the time to reach the Technology *Readiness* Level (TRL) [6]. This is because these devices provide a more relevant and accurate testing environment, allowing researchers to evaluate the efficacy and safety of medical devices at earlier stages of development [2]. As a result, new biomedical devices can be validated and optimized more efficiently, accelerating the transition to later stages of development and ultimately regulatory approval [4,7]. The ability to test medical devices in an OoC reduces the reliance on animal testing while adhering to ethical principles and strict regulations, leading to more responsible and sustainable biomedical engineering research and accelerating progress in TRLs.

Microphysiological systems are microscale models of human organs and tissues that reproduce their 3D properties and offer greater precision than conventional methods [8,9,10]. They focus on reproducing the mechanical properties of tissue and the extracellular matrix (ECM), which are essential for functionality. Microfluidics enables the creation of anatomical structures and specific organ conditions, while advances in microfabrication allow the miniaturization and integration of components such as pumps and valves. This creates more precise and physiologically relevant environments for cells, improving disease research and treatment development [9,10].

When developing these devices, it is important to take into account physical and biological factors in order to simulate realistic physiological and pathological conditions [10]. This includes biocompatibility, analysis of mechanical stresses and deformations, cell types, growth factors, cytokines and drug risks. Microfabrication plays a critical role in creating models that accurately reflect the unique characteristics of different tissues, such as size, shape, cell density and fluid flow patterns, and ensure consistency with in vivo conditions [10,11].

When creating a customized OoC platform, researchers can adjust key parameters to achieve specific study objectives. These include the selection of appropriate materials, the use of biomaterials that support cell growth and differentiation, the integration of sensors to monitor cellular parameters in real time and the ability to apply controlled mechanical forces to simulate specific physiological conditions. Rigorous validation and replication of an OoC model is an essential step to ensure the reliability and biological relevance of these tissue emulator systems [8,9,12,13]. After more than 10 years of development, OoC technology is still reaching its limits in terms of application, scalability and user-friendliness. It is estimated that the use of OoCs can reduce research, development and innovation (RDI) costs by 10–30 and is therefore seen as a promising health innovation [14].

In this systematic review, we address the design principles of OoC systems, focusing on the anatomy and physiology of organs. We examine the fabrication techniques and biomaterials commonly used to fabricate these microscale systems. We then present an analysis of recent advances in this rapidly evolving field. Our aim is to provide a comprehensive overview of microphysiological platform designs and how these advances have positively impacted preclinical research and drug discovery for human disease. The future perspectives presented in this review also emphasize the ongoing potential of biomedical engineering to further advance the development of these innovative systems for the benefit of human health. Furthermore, this work seeks to contribute meaningful insights to advance both the development and practical implementation of microfluidic devices.

## 2. Materials and Methods

### 2.1. Protocol and Registration

This systematic review has been registered in the International Prospective Register of Systematic Reviews (PROSPERO) under the number CRD42022352569 [15]. We followed the PRISMA (Preferred Reporting Items for Systematic Reviews and Meta-Analyses) guidelines for this review [16].

### 2.2. Inclusion Criteria

The inclusion criteria for this systematic review followed the PICOS (Population, Intervention, Comparison, Outcome) approach of [17]. The selected studies were concerned with the development (population) of organ-on-a-chip (OoC) models (intervention) over the last 10 years (outcome). All tests should have been performed with human tissue or primary or secondary cell lines.

### 2.3. Exclusion Criteria

We excluded studies from this review that did not meet the inclusion criteria based on the PICO approach. In particular, studies using 2D cell cultures or cultures that did not use microfluidic platforms, assays using animal donors, and in silico assays were only included if they were related to the development of the OoC platform (Appendix A).

### 2.4. Search Strategy and Information Sources

A comprehensive literature search was carried out in the Scopus, Embase, IEEE and Web of Science databases. A total of 2491 references were initially identified—Scopus (n = 17), Embase (n = 1), IEEE (n = 2471) and Web of Science (n = 2). After removing duplicates and applying a 10-year publication date filter, 1388 unique articles were selected for screening based on titles and abstracts. This process was performed in pairs using Rayyan software, following the predefined inclusion and exclusion criteria. In the second phase, the full texts were analyzed independently in pairs using a standardized data extraction form. Discrepancies were clarified by consulting a third reviewer. This procedure served to minimize bias and ensure the consistency of the selection process.

Relevant data extraction from each study was organized in a predefined table describing key points related to the characteristics of the intervention of interest, including the replicated organ or tissue model, microfabrication methodology, 3D printing materials and variables of interest related to the dimensions of the medical device (Appendix A).

### 2.5. Bias Risks and Quality in Individual Studies

Due to the heterogeneity of the interventions evaluated, it was not possible to perform a meta-analysis of all outcome data. Therefore, we analyzed the methodological quality of the studies using the Cochrane Collaboration’s RoB 2 tool adapted by the authors. We performed the assessments in duplicate and individually. A third author mediated in cases of disagreement [18].

We adapted an available version of the RoB 2 tool to create 13 domains for assessing the methodological quality of OoC studies. These domains cover various aspects, including target and hypothesis definition, objective and unbiased references, comparison with existing platform models, pressure methods and criteria, validation experiments, submission to ethics committee, microfluidic analysis, conflicts of interest, analysis of experiments to avoid bias, reproducible and standardized data, confounding factors, repetition of experiments to increase reliability and solutions or improvements that should be made.

These domains can be used to assess the risk of bias in OoC studies and determine the overall quality of the study. By carefully assessing each domain, it was possible to decide whether the risk of bias in each study was low or high.

Responses marked “LOW” indicate that sufficient evidence or appropriate measures were in place to minimize the risk of bias, suggesting that the results of the study are likely to be reliable. The term “NO INFORMATION” was used when there was insufficient data to assess the risk of bias, indicating that additional information is needed for an adequate assessment. The classification “CRITICAL” means that there is a high risk of bias that could significantly affect the validity of the study results. Similarly, studies classified as “HIGH” have a significant risk of bias and require careful interpretation of the results due to significant methodological limitations. The classification “UNCLEAR” was applied when the available data were insufficient or ambiguous so that a final judgment on the risk of bias was not possible.

The reviewers’ ratings were quantified and used to classify the studies according to the percentage of “YES” responses to the assessment criteria. Studies with ≥70% “YES” responses were classified as *high quality*, those with 50–69% as *medium quality* and studies with <50% “YES” responses as *low quality* (Appendix A).

## 3. Results

In a bibliometric review carried out in January 2023 (see Figure 1), four word groups were found with a minimum coincidence of 20 times. The results are linked to the topic of this review; the colored lines show the connections between the terms.

The aim of this systematic review was to analyze recent advances in the architecture, design and manufacturing methods of OoC devices over the last 10 years. As shown in the PRISMA flow diagram (Figure 2), a total of 9.790 articles were retrieved from the following electronic databases up to 1 June 2025: Scopus (n = 23), Embase (n = 647), IEEE (n = 2728), CINAHL/EBSCO (n = 6.382) and Web of Science (n = 05). After removing duplicates and applying a 10-year publication filter, 3.150 articles remained, which were screened independently by three pairs of authors using predefined eligibility criteria. After full-text analysis, 18 studies were selected for this review.

The analysis of the included studies revealed diverse yet complementary approaches in advancing OoC technologies. The study “Design Automation for Organs-on-Chip” proposed a computational methodology for automating the design of multiorgan systems based on physiological parameters such as tissue mass ratios and perfusion rates, enhancing standardization and predictive accuracy. Similarly, “Automated Design for Multiorgan-on-Chip Geometries” introduced a parametric workflow for generating physiologically relevant microfluidic layouts, enabling the reproducible fabrication of integrated organ systems (Figure 3).

In the experimental domain, “Innovative Stamp-Structured OoC Platform for Vascularized Tumor and Colon Models” developed a modular OoC device incorporating human cells (HUVECs, fibroblasts, A549 and Caco-2), successfully modeling vascularized tumor and intestinal barriers. The platform was validated with chemotherapeutics (doxorubicin and capecitabine), demonstrating its applicability in drug screening. Likewise, “3D-microprinted PDMS-based microfluidic vessels for Organ-on-a-Chip Applications” presented a novel method for fabricating PDMS microvessels via 3D printing. The system supported the viability of MDA-MB-231 cells and enabled tunable control of microenvironmental permeability, offering a viable approach for vascular modeling. Finally, “Assessing the Propagation of Magnetic Nanoparticles in a Microfluidic Channel and Their Behavior at the Suspension–Hydrogel Interface for On-Chip Modeling of Organs and Tissues” examined nanoparticle diffusion across hydrogel interfaces within microfluidic channels that simulate tissue environments. The study demonstrated the feasibility of using such systems for evaluating magnetically guided drug delivery strategies in OoC platforms.

The search for innovation in microfluidic devices has increased, driven by the need for more effective approaches to biomedical studies. In this systematic review, we highlight several studies that show significant advances in this field. However, the eligibility criteria can be found in the PROSPERO protocol. Few studies provided all the required information, and even those that did had limitations regarding the details of the device architecture, such as dimensions, radius and other relevant variables necessary for study replication (Table 1).

Notable contributions include Roy et al. (2015) [23], who optimized 3D design for biochips with encouraging results indicating the efficiency of 3D design. The study explores the possibilities offered by three-dimensional microfluidic devices, presenting three different architectures and highlighting the individual advantages of each approach over traditional two-dimensional layouts. Additionally, promising results are reported, including reduced contamination and improved overall routing performance when three-dimensional designs are employed. Although promising architectural advantages were reported, the study lacked quantitative metrics to support performance claims, such as flow rate consistency or fabrication precision [23].

Lee and Hong (2015) [19] developed a microfluidic chip to generate emulsion droplets, which proved to be effective in encapsulating cells. When investigating the formation process of monodisperse emulsion droplets, successful results were obtained with diameters ranging from 200 µm to 240 µm. The flow rate of the liquid decreased over time and stabilized at around 0.0375 µL/s. It was also observed that the vacuum module used stopped propelling liquids after 170 s. In addition, the ability of the microfluidic platform to encapsulate blood cells and other cells is remarkable, even given the high viscosity of mouse blood. The droplet size can be controlled by adjusting the height of the liquid in the input columns [19].

An evaluation of the effects of drugs and metabolites on HepG2 cells was proposed by [24]. They used the “body on a chip” device, which allows the integration of multiple organs on a single chip and closed circulation of the medium to study the interactions between organs and the effects of substances. The study proposed a simplified fabrication method for the device using 3D printing materials such as PDMS and photosensitive resin through a DMD-based lithography technique and process optimization [24]. Pharmacokinetics were also investigated in the study by [25], using a HepG2 liver model and A549 as the target model. The microfluidic device was fabricated by photolithography using PDMS as the material.

Another notable advance in the studies examined the migration of HeLa-GFP cells in a 3D environment in response to IL-6 at different concentrations. The chip, made with PDMS layers, examined cell movement and speed at different IL-6 concentrations. Higher concentrations accelerated cell migration, while lower concentrations reduced it. This microfluidic chip enabled effective visualization and quantification of cell movement in an in vitro 3D environment [20].

Surface micromachining to produce a miniaturized microseparator showed promising results in chemical and biological analyses by [26]. In an investigation of lab-on-a-chip manufacturing, initial tests were conducted with a prototype produced by additive manufacturing. Dimensional evaluations and practical applications were performed, demonstrating unimpeded fluid transport in the device’s channels. It was found that the geometry needed to be adjusted to avoid solution setbacks. This led to a new geometry proposal that reduced the volume of the prototype by 68%. In addition, a cost comparison showed the advantage of additive manufacturing in the production [26].

The economic viability and the need for adjustments to the geometry have been well demonstrated. The microfluidic platform developed by [27] demonstrated its ability to generate monodisperse emulsion droplets with controlled size. The flow rate and applied pressure were critical factors for the formation of these droplets. Analysis of the captured images revealed average diameters ranging from 200 µm to 240 µm. In addition, it is important to note that the platform successfully encapsulated blood cells and other cells, highlighting its potential for applications such as microencapsulation, small-scale chemical reactions and cell and particle analysis [27].

Another study investigated the intricacies of microfluidic channels with circular cross-sectional areas. They developed a Matlab (*version R2019b*), code to analyze the flow rates, taking into account variables such as channel dimensions and geometries. Using the Hardy-Cross method, they calculated the pressure losses in the system. The study assumed laminar, stable and incompressible channel geometries and investigated various configurations, including bifurcated and tree-shaped channels. Although details on flow patterns, optimal rates and uniformity were lacking, the analysis considered the geometry of the channels in terms of length, diameter and width. Using iterative methods, they investigated 2D channel networks, taking into account viscous and turbulent effects that lead to pressure losses. Their results indicate a quadratic relationship between the flow rate (Q), diameter and channel length, which was elucidated by both iterative analysis and the Hardy-Cross method [28].

An investigation was performed between endothelial and epithelial cells in a blood–air barrier under biomimetic pulsatile flow. They used human basal adenocarcinomatous alveolar epithelial cells (DsRed-A549) and human umbilical vein endothelial cells (HUVECs). The device was fabricated from PDMS using conventional soft lithography. The research investigated the frequency and magnitude of pneumatic actuation pressure in the microfluidic pump, the impact of feedback pressure on device performance and cell–cell interactions in the blood–air barrier under pulsatile flow. The results showed that the HUVECs aligned with the flow generated by the simulated heart pump. In addition, the developed blood–air barrier supports high hydrostatic pressures, and pulsatile shear stresses can influence cell alignment [21].

Hydrogels, PDMS and nano/microfabrication technologies have been used to create organoids with architecturally accurate lumens to develop physiologically relevant human tissue models [29]. Hydrogel was also investigated in the study by [30], whose aim was to create an integrated and automated system for drug screening and personalized medicine by combining different types of organoids. The researchers developed a modular microfluidic organ-on-a-chip platform in which liver organoids were generated using PHH in a GelMA solution, and heart organoids were created with standardized, aligned grooves using GelMA and hiPSC-CMs. This platform allowed continuous monitoring for up to five days and enabled large-scale drug screening [30].

Another study aimed to create a multi-well plate platform for in vitro cell cultures that allows simultaneous mechanical stimulation and electrical monitoring. As with the previously mentioned studies, different fabrication approaches were employed, with the first study utilizing polymers such as PDMS and GelMA and the second focusing on silicon wafer fabrication and encapsulation using FAM technology. In both studies, performance and biocompatibility analyses of the platforms with cell cultures were also performed to demonstrate the feasibility and usefulness of these platforms for in vitro cell studies [31].

In the work of [22], microchips were fabricated by microfabrication processes using lamination and biocompatible adhesives to create microfluidic channels that allow precise control of the flow of cell culture medium for the growth of human endothelial cells under shear stress conditions. Materials such as polyester toner, polyester film and epoxy adhesive were used to fabricate the microchips, which were previously tested for cytotoxicity and showed satisfactory results. During the experiment, the endothelial cells were cultured in a microchannel for up to 24 h, allowing for the observation of cellular morphological changes using high-resolution optical microscopy. The results showed that the microchips provided a suitable environment for cell adhesion, proliferation and cell counting [22].

These studies represent a variety of advances that point to the broad applicability and transformative potential of microfluidic devices in biomedical research. The contributions of this work are fundamental to the continued advancement of this field, fostering innovation and improving the tools available to scientists and researchers.

### Analysis of Methodological Quality in Articles

In the analysis of bias, the studies were assessed as shown in Figure 4, with six of them showing a low risk of bias [21,22,23,24,26,28]. Conversely, about three studies showed a high risk of bias [19,27,29], while four studies did not contain sufficient information or were not presented [20,25,30,31]. The lack of a clear description of the procedures used for sample selection, assignment of experimental groups and control of variables may cast doubt on the robustness and reliability of the results. The implications of this lack of methodological clarity are significant and extend to several dimensions of the research. First, the internal validity of the studies is compromised. The lack of adequate strategies to address bias can lead to biased results and incorrect conclusions. This undermines the credibility of the research and can lead to inadequate clinical or scientific recommendations.

Furthermore, the generalization of the results is also impaired. The lack of clarity in the analysis of biases makes it difficult to assess the transferability of results to other contexts or populations. The reproducibility of studies becomes a challenge when readers lack access to detailed information on bias control measures.

Another worrying implication is the missed opportunities to advance the technology of OoC devices themselves. High-quality research is crucial to drive innovation and development in this area. The lack of transparency in analyzing bias may lead to under- or overestimating the actual benefits of these devices, limiting scientific and technological progress.

Although none of the studies achieved an ideal score, the results presented here are relevant to this review. To mitigate these issues, researchers must apply more rigor when describing their methodology.

## 4. Discussion

In recent years, microfluidic devices have significantly advanced cell culture, from early experimental models in the 2000s to advances in the 2010s [11]. Three main designs illustrate this evolution: the multichannel design using hydrogels to create porous barriers that enable cellular standardization [5]; the porous membrane design that induces mechanical cell elongation through an air–liquid interface in a “lung-on-a-chip” and displays the expression of relevant lung markers [13]; and the mold strategy, which mimics 3D tubular structures for cell cultures and enables studies of angiogenesis and cell function based on geometry [32]. Our analyses relied on a wide range of sources, mainly electronic databases such as Scopus, IEEE and Web of Science. With this approach, we identified a total of 13 relevant articles for our study.

Translational research in the healthcare sector, which is concerned with the commercialization of drugs or medical devices, goes through various development phases. Conventional preclinical in vivo and in vitro trials have considerable limitations. Less than 10% of drug candidates are approved by the US Food and Drug Administration (FDA), with failures primarily related to nonclinical and clinical safety (over 50%) and efficacy (over 10%). These high failure rates are due to the inability of conventional models to predict efficacy and safety in humans [30,33]. Among the main research topics in the field of OoC devices, microfluidics stands out as an essential tool for replicating physiological and pathological conditions of human organs and tissues. The OoC approach has been explored to deepen our understanding of biological functions, to study diseases more precisely and efficiently and to enhance drug testing innovation. However, it should be noted that despite promising progress, many studies do not provide specific quantitative results, thereby limiting a comprehensive evaluation of these contributions.

Although the field of OoCs is still in its infancy, OoC models based on induced pluripotent stem cells, primary human cells and organoids are already being extensively developed. From single-organ systems to body-on-chip models, some are already being used to study diseases and identify therapeutic targets. However, there are still challenges, such as robustness, reproducibility and adaptation to existing laboratory workflows, that hinder regulatory acceptance and industry adoption as an alternative to animal models [33]. The study by [29], which looked at the production of organoids for drug screening, proposed a model that could be tested on a larger scale in the future.

Efficient design becomes one of the main topics to consider when discussing OoC development. Computer simulation is a tool that can be used to optimize processes through in silico analyses, which aid in the development of platforms. However, the use of specific quantitative data as variables is scarce [23]. Moreover, as far as OoCs for pharmacokinetic screening studies is concerned, the lack of standardization of models directly affects the translational research phases of devices [34].

Over the last two decades, the field of nanotechnology and nanomedicine has experienced explosive growth. Manipulated nanoparticles, in particular, have garnered considerable attention due to their potential to enable new possibilities, such as the controlled and targeted delivery of drugs to treat various diseases. With the rapid advances in nanoparticle research, efforts are increasingly focused on developing new technologies for in vitro modeling and analysis of the efficacy and safety of nanotherapeutics in human physiological systems [7,35]. In the study by [19], a microfluidic platform is presented that can generate monodisperse emulsion droplets with controlled size, demonstrating the feasibility of encapsulating blood cells. This study has demonstrated the practical applicability of these technologies.

In addition to efforts to make OoCs more physiologically relevant through 3D cell culture techniques, smart integration of biomaterials and microfluidic designs, much attention has been paid to the development of robust monitoring tools for these platforms [36,37]. This review aims to present, for the first time, a comprehensive analysis of existing methods for monitoring OoCs [38]. By examining physical, chemical and biochemical sensors used in these systems and cell culture monitoring, this work critically evaluates the advantages and disadvantages of each method, contributing to a deeper understanding of monitoring strategies in this innovative field.

In situ analysis and the behavior of cell movements in the 3D environment are extremely important for the visualization and quantification of these physiological effects [20]. The integration of miniaturized microseparators for biochemical and pharmacological analyses underscores the effectiveness of additive manufacturing in device production [26]. “Studies on Biological Barriers Under Pulsatile Flow” [21] investigated cell–cell interactions in blood–air barriers and showed the influence of pulsatile flow on cell orientation. Using an innovative biomaterial, microchips for cell growth under shear stress were developed [22], which control the flow of the cell culture medium and thus enable studies under shear stress conditions. Electrical monitoring and mechanical stimulation were also the focus of the study in [31], which cultured cells from cervical cancer patients.

In terms of practical applications, our review has revealed a wide range of promising scenarios. From the detection of pathogens to the cultivation of germ cells and the analysis of organoids, OoC devices have proven their potential in various fields. The integration of sensors, process automation and the ability to monitor cellular parameters in real time has contributed to the expansion of these applications. Despite these advances, it is noteworthy that not all studies provided quantitative empirical results or comprehensive analyses. Many provided notable contributions on approaches, methods and challenges; however, the lack of concrete data may limit a full assessment of the research’s impact.

The included studies illustrate a progressive maturation of OoC technologies, ranging from computational design tools to functionally validated microfluidic models. The computational frameworks presented by “Design Automation for OoCs” [39] and “Automated Design for Multiorgan-on-Chip Geometries” [40] highlight the growing importance of in silico approaches for scaling and optimizing OoC architectures. By incorporating physiological variables such as tissue mass and perfusion dynamics, these studies address a critical barrier in the field: the lack of standardized design criteria that ensure biological relevance and interorgan integration.

Experimental contributions further strengthen the field by demonstrating practical implementations of OoC systems using human-derived cell lines and advanced microfabrication. “Innovative Stamp-Structured Organ-on-a-Chip Platform for Vascularized Tumor and Colon Models” [41] is particularly noteworthy for integrating vascularized tumor and intestinal models, demonstrating drug responsiveness and underscoring the potential of OoCs in oncology research and personalized medicine. Similarly, the 3D microprinted PDMS vessels reported in “3D-Microprinted PDMS-Based Microfluidic Vessels for Organ-on-a-Chip Applications” [42] represent a key advance in replicating microvascular complexity, achieving the structural fidelity and biological compatibility essential for real-time tissue modeling.

Moreover, the investigation of magnetic nanoparticle behavior within hydrogel interfaces, as described in “Assessing the Propagation of Magnetic Nanoparticles in a Microfluidic Channel and their Behavior at the Suspension–Hydrogel Interface for On-Chip Modeling of Organs and Tissues” [43], provides valuable insights into the feasibility of controlled drug delivery in on-chip environments. The study highlights the importance of mechanical and diffusion properties of biomimetic materials in achieving precise targeting and interaction with engineered tissues.

Collectively, these studies reveal that organ-on-a-chip systems are moving toward greater physiological realism, modular integration and functional versatility. However, challenges remain regarding the translation of these models into standardized platforms for regulatory acceptance, particularly in drug discovery pipelines. Future work should focus on comparative validation against in vivo data and expansion of multiorgan interactions under dynamic conditions.

## 5. Conclusions

In the field of microfluidics, OoC designs have proven to be a promising approach to mimic physiological and pathological conditions of human organs and tissues in a controlled microscopic environment. These miniaturized devices have been extensively studied to better understand complex biological functions, investigate diseases with greater precision and efficiency and enable innovative, controlled drug testing.

The use of biomaterials plays a crucial role in the construction of OoC devices and is as essential an element as the design. Standardization in the selection and use of these materials is necessary to ensure the reproducibility and reliability of the results obtained. Additionally, the possibility of large-scale production is a crucial factor for the future commercialization of these devices. The ability to mass-produce these devices not only ensures consistent quality but also makes these technologies more accessible, opening the doors for wide application in research and industry. This standardization and feasibility of mass production can be crucial for the successful commercialization and wide acceptance of these devices in the market.

In this article, our review culminates in an in-depth analysis of the main findings of recent research in this field, highlighting the significant contributions of these advances. The review encompasses various facets, ranging from the fabrication of microchannels and culture chambers utilizing technologies such as 3D printing to the development of microfluidic platforms equipped with sensors that can monitor soluble biomarkers, microenvironmental parameters and organoid behavior in real time.

However, it is essential to note that while we reviewed various studies, many did not provide specific or quantitative empirical results. The information about the architecture of the devices was often not well specified, which is the major limitation of this study.

## Figures and Tables

**Figure 1 biomimetics-10-00524-f001:**
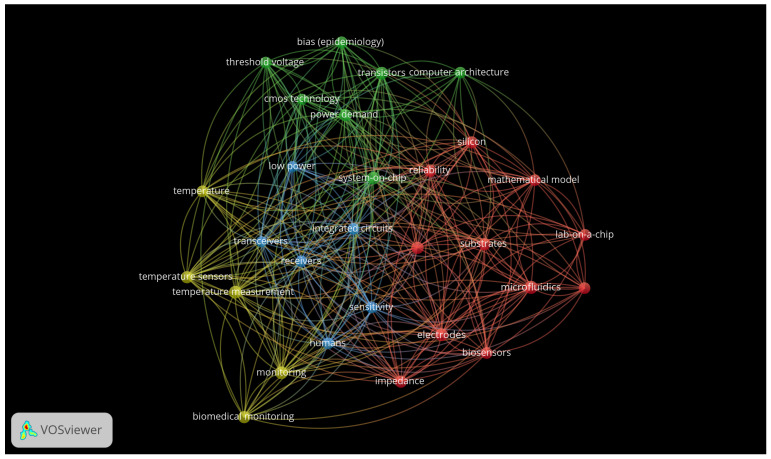
Bibliometric analysis: Word map—The bibliometric analysis was carried out with the program VOSviewer Version 1.6.17 on 1.276 works that contained the search terms (“Organ on a chip” OR “Lab-on-a-chip device” OR “Lab on a chip” OR “Body on a chip” OR “Three Dimensional Cell Culture” AND “Patient-Specific Modeling” OR “Cellular Microenviroment” OR “Computer Simulation”AND “Bioprinting” OR “Computer Aided Design” OR “Equipment Design” AND “Biomedical Technology”) in the title or keywords on 24 July 2023. The analysis was conducted using a minimum coincidence of terms of 20 times and binary counting.

**Figure 2 biomimetics-10-00524-f002:**
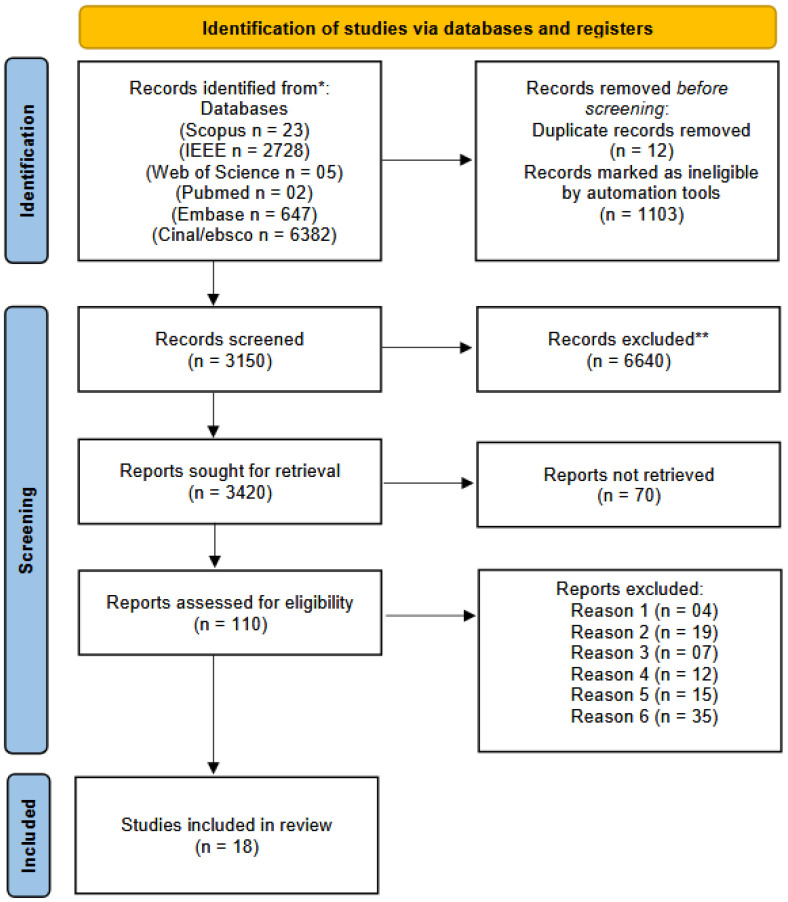
Flow diagram of literature search and selection criteria adapted from PRISMA [16]. Source: authors. * Consider, if feasible to do so, reporting the number of records identified from each database or register searched (rather than the total number across all databases/registers). ** If automation tools were used, indicate how many records were excluded by a human and how many were excluded by automation tools.

**Figure 3 biomimetics-10-00524-f003:**
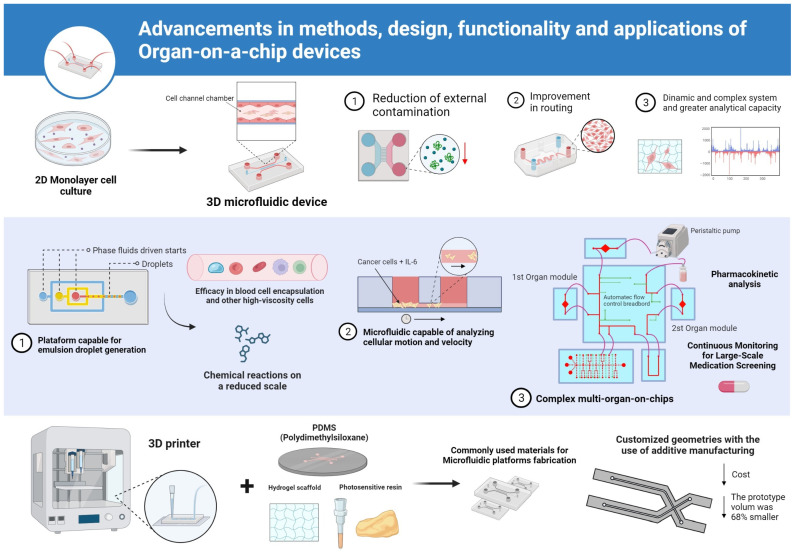
Advancements in methods, design, functionality and applications of organ-on-a-chip (OoC) devices. (1) Transition from conventional 2D monolayer cell culture to 3D microfluidic platforms with perfusion channels (typical channel width: 100–500 µm), allowing improved routing and reduced external contamination. (2) Development of specialized microfluidic devices for high-viscosity droplet generation (e.g., 200–240 µm in diameter), blood cell encapsulation and real-time analysis of cancer cell migration velocity under IL-6 stimuli. (3) Integration of multiorgan modules and peristaltic pumps enabling pharmacokinetic analysis and continuous monitoring of large-scale drug screening (typical flow rate: 0.01–1 µL/s). Additive manufacturing techniques using materials such as PDMS, hydrogel scaffolds and photosensitive resin allow custom geometries and significant volume reduction (e.g., from 1.2 cm^3^ to 0.38 cm^3^; 68% decrease). Schematic representations are illustrative and not to scale.

**Figure 4 biomimetics-10-00524-f004:**
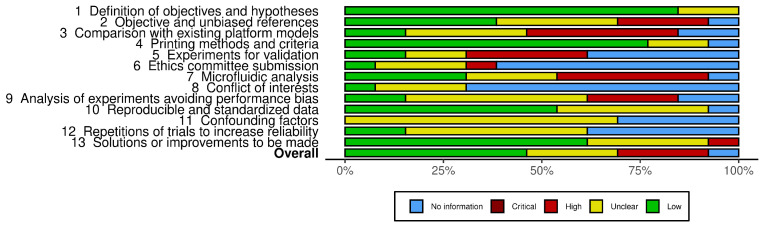
Overall quality of selected studies.

**Table 1 biomimetics-10-00524-t001:** Summary of available quantitative parameters reported in selected studies. Only articles that explicitly presented measurable data (e.g., droplet size, flow rate, shear stress) are included. “*Not reported*” indicates that the information was not available in the original publication, reflecting the heterogeneity and incomplete reporting in the field.

Study	Device Type	Droplet Diameter (µm)	Flow Rate (µL/s)	Cell Type	Shear Stress (dyn/cm^2^)	Culture Duration (h)
Lee & Hong, 2015 [19]	Droplet generator chip	200–240	≈0.0375	Blood cells	*Not reported*	*Not reported*
Yeh et al., 2019 [20]	3D migration chip	*Not reported*	*Not reported*	HeLa-GFP	Variable (IL-6-induced)	*Not reported*
Ko et al., 2020 [21]	Blood–air barrier chip	*Not reported*	Simulated (pulsatile)	A549 + HUVEC	Pulsatile flow	24
Ma et al., 2021 [22]	Shear stress microchip	*Not reported*	Controlled	Endothelial	Present	24
Roy et al., 2015 [23]	3D microfluidic biochip	*Not reported*	*Not reported*	Not specified	*Not reported*	*Not reported*

## Data Availability

The data sets used and/or analyzed in this study are available on request from the corresponding author. For further inquiries, please contact the corresponding author.

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
