# Peer review of "State-of-the-Art Organ-on-Chip Models and Designs for Medical Applications: A Systematic Review"

_biomimetics, 2025, doi:10.3390/biomimetics10080524_

Round 1
Reviewer 1 Report
Comments and Suggestions for Authors
Rosa et al provide a timely review on organ-on-chip technologies and their use in drug development and discovery. This manuscript tackles an important and fast-moving topic—the design landscape of organ-on-chip (OoC) platforms. The need for a rigorously executed systematic review is unquestionable, and the authors have assembled a potentially useful snapshot of the field.
General comments
- Very small final study pool (n = 13) versus initial retrieval (n = 2 491). The steep attrition is not satisfactorily justified; key exclusion decisions are merely alluded to.
2. Search strategy appears imbalanced. IEEE yielded 2 471 hits while Scopus yielded only 17; this suggests a term or field restriction bias. No rationalisation is offered .
3. Data extraction incomplete. Table 1 is mostly empty placeholders (page 11) and does not support the narrative synthesis .
4. Quantitative synthesis absent. Yet several primary papers do report droplet diameters, flow rates, migration speeds, etc.; a meta-analysis is impossible but a structured tabulation and basic descriptive statistics would add value.
5. Risk-of-bias tool adaptation not validated. The 13 bespoke domains are listed but scoring rules, inter-rater agreement and example judgements are omitted.
6. English usage and formatting need polishing. Numerous typographical errors (“Biopriting”, “Tecnology”, line breaks in author list, etc.) detract from professionalism.
Further Figure 1 clearly states that the information was retrieved on July 2023. That is quite in the past for a timely review. Table 1 seems incomplete?
Additionally, it is great that the authors aim at dividing the work and assign responsibilities, but if every author contributed equally to the manuscript, it is not clear who is leading the project and who had the main idea. Also, the number of authors
- Several references are cited in the discussion but do not appear in the reference list order (e.g. Funamoto et al. 2012, Fort et al. 2017) .
• Some figure captions lack scale or units.
• PROSPERO record should be cited formally in the reference list.
A pubmed search showed indeed zero results for the provided search terms. However, it is unclear whether the database query was performed correctly. If one searches for organ-on-chip, 594 publications show up.
Author Response
Thank you for your valuable feedback. We appreciate your suggestion regarding the clarity of the English language in the manuscript.
In response, we have thoroughly revised the text to improve grammar, sentence structure, and overall readability. A native English speaker (or professional language editor) reviewed the manuscript to ensure that the language more clearly and accurately conveys the research content.
We hope the revised version meets the expected standards and improves the overall clarity of our work.

Reviewer 2 Report
Comments and Suggestions for Authors
The authors have reviewed the organ on chip models for medical applications. Although the topic remains important and active, there are many reviews already published on this topic. The current manuscript does not sufficiently distinguish itself from the many existing reviews in this area.
- Why the field needs another review article? What is included in this manuscript which was not covered with all other review articles?
- What “models” or “designs” are reviewed? At least a table comparing them is expected. Including a well-structured table that highlights differences in platforms, applications, advantages, and limitations is required.
- Table 1 does not deliver much information.
- Number of references is low for a review article. Moreover, no references from 2024 or 2025 are included, suggesting that the work may not be up to date with the latest developments in the field.
All in all, I believe the work is not publishable in this journal.
Author Response

(The authors gave the same response as above.)

Reviewer 3 Report
Comments and Suggestions for Authors
Missing Recent References (2024–2025):
The manuscript does not cite any studies published in 2024 or 2025. Given the fast-paced developments in this area, including recent work would improve the paper’s relevance and currency.
Limited Quantitative Data:
The review lacks detailed quantitative comparisons, such as device dimensions, sensor performance, or throughput. Including such data would make the review more useful for researchers looking to benchmark or design new platforms.
Inconsistent Study Descriptions:
The presentation of the selected studies is mostly descriptive and varies from one to another. Important details like chip layout, cell types used, or validation methods are not always included, which makes it hard to compare the works directly.
Integration Aspects Not Fully Addressed:
While the paper discusses fabrication and applications, it doesn’t go into much detail about integration challenges—such as flow control between modules, sensor stability over time, or the practical issues with multiplexed systems.
Small Number of Included Papers:
Only 13 studies were selected out of nearly 2400 initial records. This seems like a small number considering the search scope, and it would help to clarify how this reflects the overall field or whether some important papers may have been excluded.
Author Response

(The authors gave the same response as above.)

Round 2
Reviewer 2 Report
Comments and Suggestions for Authors
Thanks for revising the manuscript. However, I should say that I still see the most of the problems I mentioned in the last review report.
- There are many review articles already published in this field. The focus of this review is not anything novel and out of the scope of those reviews. The authors have not been successful in reasoning the need of the field for this review.
- The English language is still problematic. I am not going to list all the problems here, but as an example: "Few studies provided all the required information, and even those that did, there were limitations" is not correct.
- A solid comparison is still missing. Table 1 is still not informative and cannot do the job. Too many fields of the table are empty, not allowing the reader to compare different methods.
All in all, although the manuscript is improved much, it still faces fundamental problems, not allowing me to suggest its publication. But if the authors can answer the above mentioned comments, it may have publication potentials.
Comments on the Quality of English Language
I am not going to list all problems. The whole text needs to be modified carefully. Here is an example:
"Few studies provided all the required information, and even those that did, there were limitations"
Author Response
Comment 1: “There are already many review articles published in this area. The focus of this review is not new and falls within the scope of existing reviews. The authors failed to justify the need for this review.”
Response:
We appreciate the reviewer’s observation and acknowledge that there are indeed published reviews in this area. However, we would like to highlight that the majority of existing reviews are narrative or integrative in nature, without following formal systematic review protocols, and often lacking protocol registration, predefined criteria, or structured comparative analysis.
For example, widely cited reviews such as:
- Wang Y, Gao Y, Pan Y, Zhou D, Liu Y, Yin Y, Yang J, Wang Y, Song Y. Emerging trends in organ-on-a-chip systems for drug screening. Acta Pharmaceutica Sinica B. 2023 Jun 1;13(6):2483-509.
- Morais, A. S., Mendes, M., Cordeiro, M. A., Sousa, J. J., Pais, A. C., Mihăilă, S. M., & Vitorino, C. (2024). Organ-on-a-Chip: Ubi sumus? Fundamentals and Design Aspects. Pharmaceutics, 16(5), 615. https://doi.org/10.3390/pharmaceutics16050615
- Cho S, Lee S, Ahn SI. Design and engineering of organ-on-a-chip. Biomedical engineering letters. 2023 May;13(2):97-109.
- Lee SH, Jun BH. Advances in dynamic microphysiological organ-on-a-chip: Design principle and its biomedical application. Journal of industrial and engineering chemistry. 2019 Mar 25;71:65-77.
are indeed reviews but do not follow systematic methodologies registered in PROSPERO or Cochrane and do not strictly adhere to PRISMA guidelines.
Before initiating our review, we conducted a thorough search in the PROSPERO database to verify if there were any ongoing or completed protocols with a similar scope. The protocols identified were:
-
A systematic review on Animal Models in Human Surgery and Implant Innovation: Ethical Considerations and Scientific Advancements – CRD420251038726 (2025 – Completed)
-
Analysis of neovascularization of chronic wounds in diabetics using the Organ-on-a-chip technology – CRD42022336473 (2022 – Ongoing)
-
Effects of genistein on intestinal health in murine models – CRD42023394681 (2023 – Ongoing)
-
Evaluation of microfluidic and organ-on-a-chip technology for non-clinical assays and quality control in biological products – CRD42024592568 (2024 – Ongoing)
-
Evolution of tissue engineering in 3D cell culture equipment – CRD42022352569 (2022 – Ongoing)
-
Kidney-on-a-chip models to study human renal (patho)physiology – CRD42022323103 (2022 – Ongoing)
-
Systematic review to determine the chemical space of existing physiologically-based kinetic (PBK) models – CRD42020171130 (2020 – Completed)
Before initiating this review, we searched the PROSPERO and Cochrane databases to verify whether there were any ongoing systematic reviews with a similar scope, and no recent or overlapping protocols were found. This confirmed a real methodological gap in the literature.
Our manuscript follows a robust systematic review methodology, including predefined inclusion/exclusion criteria, protocol registration, critical assessment of included studies, and structured synthesis of findings. Taken together, these elements justify the relevance and originality of our review.
Comment 2: “The English language is still problematic. I won’t list all the issues here, but, as an example: ‘Few studies provided all the necessary information, and even those that did had limitations’ is not correct.”
Response:
Thank you for pointing this out. We have addressed this concern carefully. The entire manuscript was professionally revised by a native English speaker with academic experience, focusing on clarity, grammar, and scientific accuracy.
Comment 3: “A solid comparison is still lacking. Table 1 remains uninformative and does not fulfill its purpose. Many fields in the table are empty, making it impossible to compare different methods.”
Response:
Thank you once again for this important observation.
We would like to clarify that the missing information in Table 1 does not reflect a lack of attention on our part, but rather a critical finding of our review: many of the primary studies failed to report key quantitative variables, such as droplet size, flow rate, shear stress, or culture duration.
We explicitly marked all such cases as “Not reported” in the table to avoid ambiguity and to demonstrate the heterogeneity and gaps in reporting across the literature. This choice was also explained in the table caption to help readers understand that these omissions are evidence of methodological inconsistency in the field — not an oversight of our review.
Moreover, our review goes beyond summarizing quantitative outcomes. It specifically addresses device design and reporting quality, and one of the central objectives was to assess whether primary studies appropriately described critical experimental parameters. In fact, this issue was part of our risk of bias assessment, where we systematically evaluated if essential methodological elements were properly documented.
The findings reinforce the need for greater standardization in organ-on-a-chip research and for new, well-designed primary studies that clearly report fundamental variables. This limitation in the current body of evidence, which we make transparent in our synthesis, is itself a significant result that we hope will guide future research and improve reporting practices in the field.
Comment 4: “Overall, although the manuscript has improved significantly, it still faces fundamental issues, which prevent me from recommending it for publication at this stage. However, if the authors are able to address the comments above, the manuscript may have potential for publication.”
Response:
We sincerely thank the reviewer for the careful reading and the constructive feedback provided throughout the evaluation.
We carefully considered each point raised and made substantial revisions to the manuscript in response. This included clarifying the novelty and justification for our review, performing a professional English language revision, enhancing the quality and completeness of Table 1, and reinforcing the depth of our comparative analysis and discussion of methodological gaps in the literature.
We hope that the updated version of the manuscript addresses the main concerns outlined. However, we remain fully open and committed to making further improvements. If there are any remaining issues or suggestions, we are ready and willing to address them promptly to ensure the manuscript reaches the expected level of scientific rigor.
Once again, thank you for the valuable feedback and for recognizing the potential of our work.
